# Changes in the Raman Spectrum of Monolayer Graphene under Compression/Stretching Strain in Graphene/Piezoelectric Crystal Structures

**DOI:** 10.3390/nano13020350

**Published:** 2023-01-14

**Authors:** Artemy Irzhak, Dmitry Irzhak, Oleg Kononenko, Kirill Pundikov, Dmitry Roshchupkin

**Affiliations:** Institute of Microelectronics Technology and High Purity Materials, Russian Academy of Sciences, Chernogolovka, 142432 Moscow, Russia

**Keywords:** graphene, LiNbO_3_ domain structure, PCA, Raman spectroscopy

## Abstract

Results from studying the effect of an applied electric voltage on the Raman spectrum of graphene deposited on a lithium niobate crystal substrate with a ferroelectric domain structure are presented. The use of the principal component method for data processing in combination with correlation analysis made it possible to reveal the contribution to the change in the spectra associated with the linear deformation of the substrate due to the inverse piezoelectric effect. An effect of the graphene coating peeling was found. Furthermore, bending deformations of the graphene coating associated with the presence of a relief on the substrate were found. An analysis of the change in the spectra of graphene under the application of an electric voltage made it possible to determine the height of this relief.

## 1. Introduction

The integration of the properties of traditional acoustic materials, such as piezoelectric single crystals with piezoceramic 1D and 2D films, opens up new opportunities for creating hybrid acoustoelectronic devices and various sensor devices based on them [1]. We are especially interested in the use of piezoelectric single crystals with good piezoelectric properties (LiNbO_3_, LiTaO_3_, La_3_Ga_5_SiO_14_) in hybrid devices [2,3,4,5]. In the domain of 2D materials, the use of graphene, a material with unique physical properties that has good conductivity and is transparent in the optical wavelength range, is of particular interest [6]. The combination of graphene with other materials makes it possible to solve important problems, for example, in solar cells, where a significant improvement in efficiency can be achieved by increasing the working surface as a function of the transparency of conducting graphene electrodes [7]. The development of solar energy is also associated with the development of the acoustically stimulated transport of charge carriers, where a surface acoustic wave allows the collection of a large number of charges from the surface [8,9]. Recently, the study of acoustic wave propagation in piezoelectric crystals with a graphene film formed at the surface has received considerable attention. In this case, the acoustically stimulated transport of charge carriers is also studied since the transport of charge carriers is determined both by the potential applied to the graphene film and by the amplitude of the surface acoustic wave [10,11,12,13]. Moreover, graphene can be used directly to fabricate interdigital transducer structures for the excitation of surface acoustic waves [14].

Further investigation of the characteristics of graphene/piezoelectric structures will make it possible to expand their application in sensorics, acousto-optics and optoelectronics. An important direction is the development of non-destructive methods for characterizing both the substrate material and the 2D coating itself (graphene). One attractive technique is Raman spectroscopy. The main problem of this method is the noise of the spectra, as a result of which it is very challenging to reveal the influence of any small effects in the structures under study on the Raman spectra. This work presents results from a study on the possibility of using the principal component method in combination with correlation analysis to analyze the effect of piezoelectric substrate deformation on the coating of monolayer graphene. Previously, this approach has been used to probe changes in the composition and piezoelectric properties of crystals [15,16].

## 2. Experimental Set-Up

To study the effect of compression/stretching strain on the Raman spectra of monolayer graphene, a sample was fabricated based on a Z-cut substrate of a lithium niobate single crystal with a ferroelectric domain structure. The domain width is 500 µm for a sample thickness of 0.5 mm.

On the upper face of the substrate, a monolayer graphene coating is formed by the transfer method. This coating serves as the upper electrode. A monolayer CVD graphene on a copper foil manufactured by Rusgrafen was used. The transfer is carried out as follows. A polymethyl methacrylate (PMMA) film is deposited onto a copper foil with grown monolayer graphene by spin coating. After drying the PMMA, the graphene is removed from the backside of the copper foil in oxygen plasma. Then, the foil is etched in a solution of ferric chloride. After that, PMMA with graphene is washed in deionized water and caught on a LiNbO_3_ substrate. An aluminum electrode is deposited on the underside of the substrate. The application of electrical voltage to the top and bottom electrodes causes periodic deformation of the substrate due to the inverse piezoelectric effect. The sign of deformation is determined by the direction of the crystallographic Z axis of the LiNbO_3_ substrate. A schematic representation of the deformed sample is shown in Figure 1.

The deformation of the graphene coating occurs due to the deformation of the substrate and can be studied using Raman spectroscopy. For this purpose, a Bruker Senterra Raman microscope with a 50× objective is used. The laser wavelength is 532 nm, with a power of 10 mW. The laser beam diameter is 2 μm. Electrical voltage is applied in the range from −1500 V to +1500 V with a step of 250 V. Thus, the maximum electric field strength applied to the sample is ±3 kV/mm, which is far from the values at which nonlinear effects appear [17].

## 3. Experimental Results

Figure 2 shows a set of experimental Raman spectra obtained with a cyclic application of an electrical voltage in the range indicated above. For a level of statistical significance at α < 0.05, a power of correlation test (1 − *β*) = 0.9 and a correlation factor |r| ≥0.5, the sample size should be at least *N* = 29. A total of 73 spectra was obtained, which corresponds to a total of three cycles of changes in the applied electrical voltage. Thus, the analysis of the results should be counted as statistically reliable and will make it possible to establish the presence of a correlation between the changes in the spectra and the applied electrical voltage.

The spectra contain peaks characteristic of monolayer graphene, since the full width at half the maximum of the 2D and G line peaks is less than 40 cm^−1^, and the I(2D)/I(G) ratio is greater than 1 [18]. There is also a lithium niobate peak and a peak that is presumably associated with nitrogen adsorbed on graphene. This assumption is supported by the presence of a weak peak to the left of the G-band of graphene, which, apparently, is associated with oxygen. The position, as well as the ratio of the intensities of the oxygen and nitrogen lines [19,20], indicates a high probability that the assumption is correct. There is only one mention in the literature of the observation of atmospheric nitrogen lines in the spectra of carbon materials: carbon nanofibers, carbon black and graphene powder [21,22]. Thus, this is the first observation of the presence of atmospheric air spectral lines in the Raman spectra of monolayer graphene. A possible reason for the appearance of peaks of atmospheric gases is the combination of graphene, the polar dielectric substrate, and the presence of a relief on the surface of the substrate.

It should be noted that a significant spread of the background of the obtained spectra was observed, as well as the noisiness of the peaks. In addition, it is known that the dependence of the position and intensity of the G-peak of graphene, depending on the applied electric field, is nonlinear [23]. The combination of these factors significantly complicates the analysis of the changes in intensity and position under the influence of substrate deformation caused by the application of an electrical voltage. To isolate the changes in the spectra caused by the action of the applied electrical voltage, the principal component (PC) method is used in the analysis of the spectrometric data in combination with correlation analysis. The essence of the approach is that, after decomposing the matrix of experimental data into principal components, the correlation of the principal components with the dependence of the change in the applied electrical voltage is investigated. Based on the geometric meaning of the principal component method (the transition of the matrix to the coordinate system associated with the directions in which the largest data change is observed), the presence of a statistically significant correlation of the change in the electrical voltage applied to the sample with any of the principal components will mean that the change in the electrical voltage affects the Raman spectrum. The nature of the change in the spectrum will be determined by the load corresponding to the main component.

Figure 3 shows the dependence of the correlation factor *r* on the number of the main component.

The correlation coefficient of two random variables is a measure of their linear dependence. If each variable has *N* scalar observations, then the Pearson correlation coefficient is defined as [24]
$$r(A,B)=\frac{1}{N-1}\sum_{i=1}^{N}\left(\frac{A_{i}-\mu_{A}}{\sigma_{A}}\right)\cdot \left(\frac{B_{i}-\mu_{B}}{\sigma_{B}}\right)$$
where *μ_A_* and *σ_A_* are the mean and standard deviation of *A*, respectively, and *μ_B_* and *σ_B_* are the mean and standard deviation of *B*.

The range of *p*-values lies between 0 and 1, where values close to 0 correspond to a significant correlation in *r* and a low probability of observing the null hypothesis. The *p*-values are calculated from the deviation between the observed value and a chosen reference value, given the probability distribution of the statistic, with a greater difference between the two values corresponding to a lower *p*-value. Mathematically, the *p*-value is calculated using integral calculus from the area under the probability distribution curve for all values of statistics that are at least as far from the reference value as the observed value is relative to the total area under the probability distribution curve.

A correlation corresponding to the above parameters of statistical significance is observed for the seventh principal component PC7: r = 0.62, *p* = 6.05 × 10^−6^.

The linear regressions of PC7 for the full data set and for each cycle separately are shown in Figure 4. An analysis of the results shows that, in each cycle, the correlation of PC7 with the change in the applied electrical voltage is stronger than for the full data set (Table 1). This is due to the change in the parameters of linear regression for each cycle.

Figure 5 shows the dependences of the regression coefficients on the cycle number of the change in the applied voltage. It can be seen that while the coefficient at the variable representing the change in the applied voltage (slope) decreases, the free term of the expression describing the linear dependence (intercept) increases, changing from negative values for the first cycle to positive ones for the third cycle. This indicates that the characteristics of the structure under study change during measurement.

A comparison of the load vector for the first principal component, which is essentially a spectrum cleared of changes caused by any influences, with the load vector for the seventh principal component, for which a linear dependence on the applied electrical voltage was found, is shown in Figure 6. It can be seen that for the 2E peak of lithium niobate, a shift in the position of the maximum is observed, depending on the sign and magnitude of the applied electric field.

A periodical change in the intensity of the peak presumably associated with nitrogen adsorbed (N_2_ in Figure 2 and Figure 6) on the graphene surface is also observed, which indicates a change in the concentration of adsorbed atoms with a change of the applied voltage. This phenomenon can be used in nitrogen gas sensors.

Most interesting is the fact that the graphene peaks shifted from the position of the maximum of the main graphene peaks, showing a linear dependence on the magnitude of the applied electrical voltage. Thus, next to the G peak, a satellite is found shifted by 11 cm^−1^ towards larger wave numbers, which changes the position of the maximum in the range ±5 cm^−1^, depending on the magnitude and sign of the applied electric field. The characteristic shape of the peak, shifted by 6 cm^−1^ towards lower wave numbers from the 2D peak, indicates a change in its amplitude and width at half height.

Obviously, the change in the position of the 2E peak is associated with the inverse piezoelectric effect. The combination of changes in the peaks next to G and 2D can be explained by the presence of bending deformations of the graphene coating [25].

Based on the above analysis, a conclusion can be drawn about the structure of the sample under study. First, the domain structure initially had a relief that caused the bending of the graphene coating, which manifested itself in the splitting of the G and 2D peaks. The application of voltage caused a change in the height of the relief. Based on the fact that the change in the position of the peak did not reach the main peak of graphene, it can be concluded that the relief did not disappear during the change in voltage. This means that the initial height of the relief was greater than its maximum change due to the application of an electric field. The height of the initial relief can be estimated according to the following reasoning: as a result of the application of an electric field, one part of the crystal is compressed by *δh* according to the inverse piezoelectric effect equation for the given experimental conditions:*δh* = V∙*d*_33_,
where *δh*—crystal size changing along voltage applying direction; *V*—applied voltage; *d*_33_—piezoelectric strain coefficient of the LiNbO_3_ crystal. Piezoelectric constants of LiNbO3 can be found in [26], such that *d*_33_ = 6.2 pC/N. A simple calculation yields the change of crystal size for the applied voltage as V = 1500 V *δh =* 9.3 nm. The second part of the crystal is stretched by the same amount.

In the case of increasing relief height, a shift of the peak by 5 cm^−1^ towards larger wave numbers is observed. The peak shifts towards lower wave numbers when the relief height decreases by the same value. It is shown above that the dependence of the peak position on the applied voltage is linear. Thus, in order for the peak to coincide with the main graphene peak G, it is necessary to apply approximately twice as much voltage, i.e., 3000 V. Consequently, the height of the domain structure relief without the application of an electric field is 37.2 nm.

## 4. Conclusions

As a result of this research, a linear dependence of the Raman spectrum of graphene deposited on the domain structure of a lithium niobate crystal on the applied electric voltage was isolated. This part of the contribution to the change in the Raman spectrum is associated with the deformation of the piezoelectric substrate due to the inverse piezoelectric effect. It was found that, during the cyclic application of an electric field, the effect of substrate deformation on the change in the Raman spectrum weakens from cycle to cycle. This may be due to the peeling of the graphene coating. It was also found that the substrate has a relief due to the presence of a ferroelectric domain structure.

## Figures and Tables

**Figure 1 nanomaterials-13-00350-f001:**
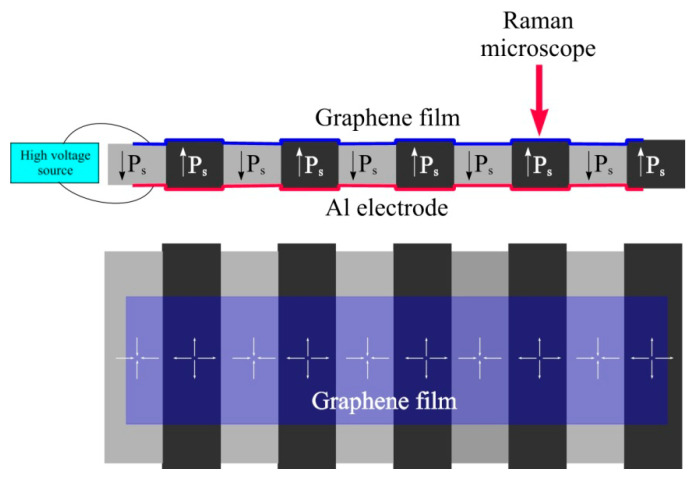
Schematic representation of the sample (cross-section and top-down view).

**Figure 2 nanomaterials-13-00350-f002:**
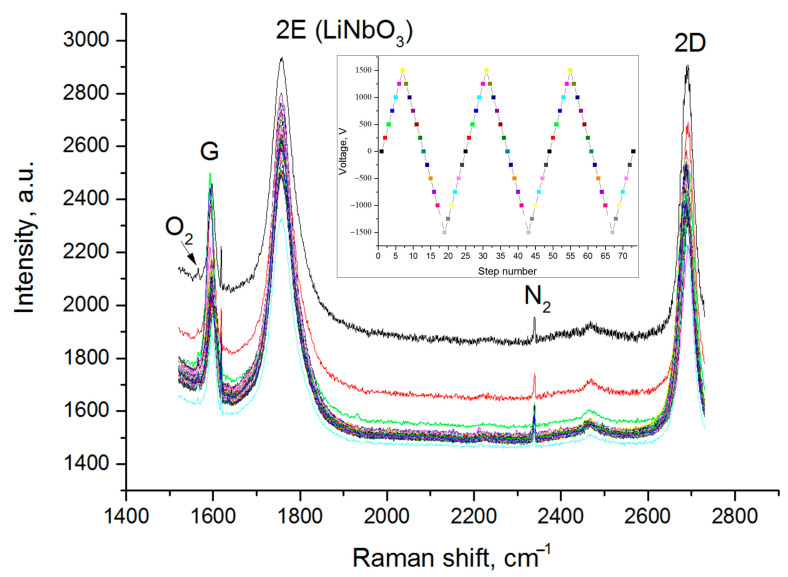
Experimental Raman spectra from graphene on the substrate of a Z-cut lithium niobate single crystal, obtained at various electrical voltages (peak near G: instrument artifact). The inset shows the change in applied voltage. The color of the dots in the inset corresponds to the color of the spectrum in the main plot.

**Figure 3 nanomaterials-13-00350-f003:**
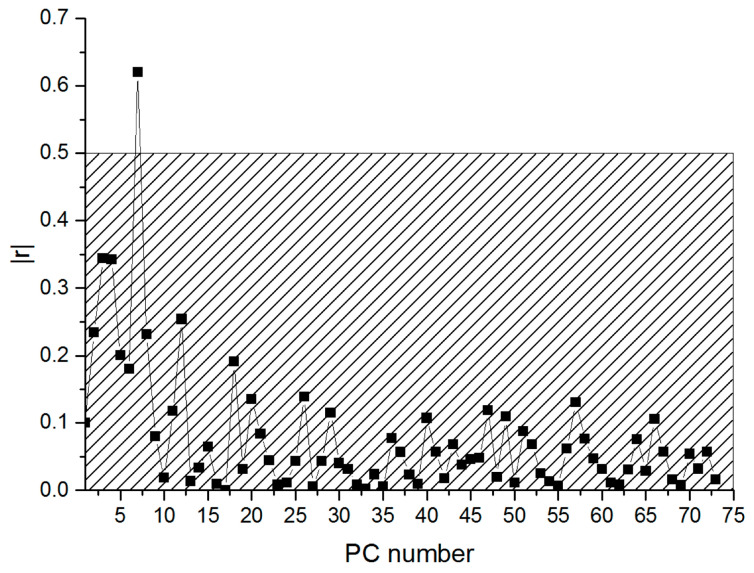
Correlation factor *r* for various principal components.

**Figure 4 nanomaterials-13-00350-f004:**
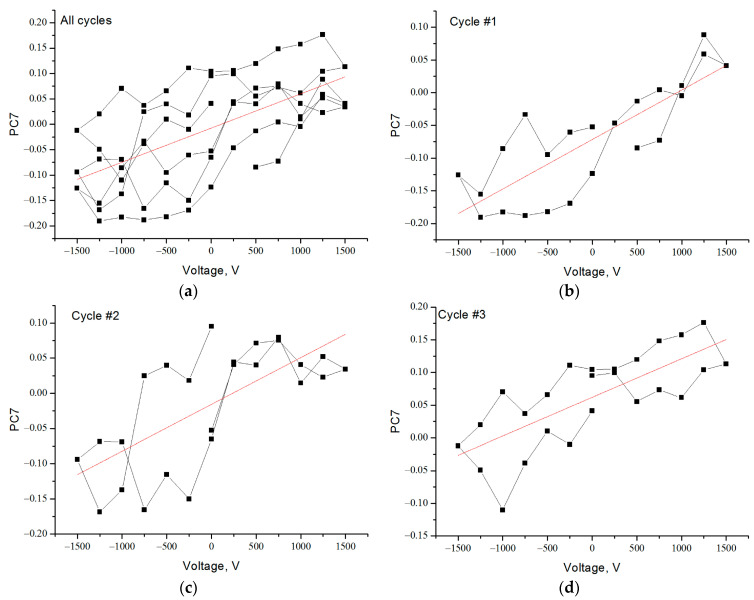
Dependences of the PC7 value on the applied voltage (black line) and the corresponding linear regressions (red line): (**a**) for all cycles of voltage change; (**b**) for the first cycle; (**c**) for the second cycle; (**d**) for the third cycle.

**Figure 5 nanomaterials-13-00350-f005:**
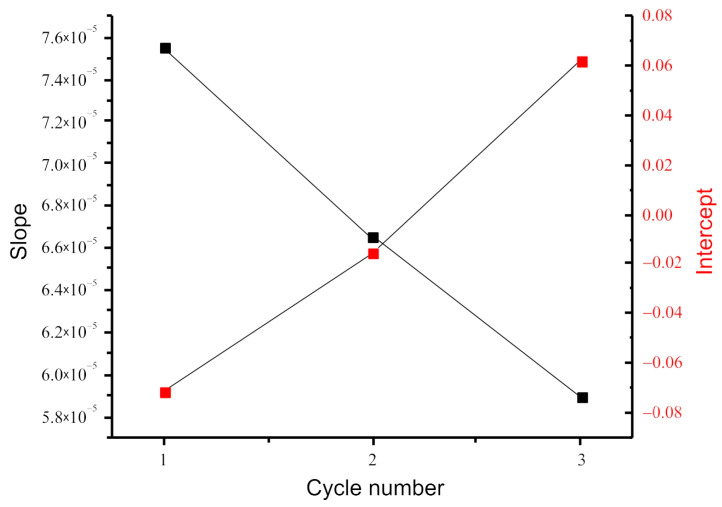
Dependences of the parameters of linear regressions on the cycle number of voltage change.

**Figure 6 nanomaterials-13-00350-f006:**
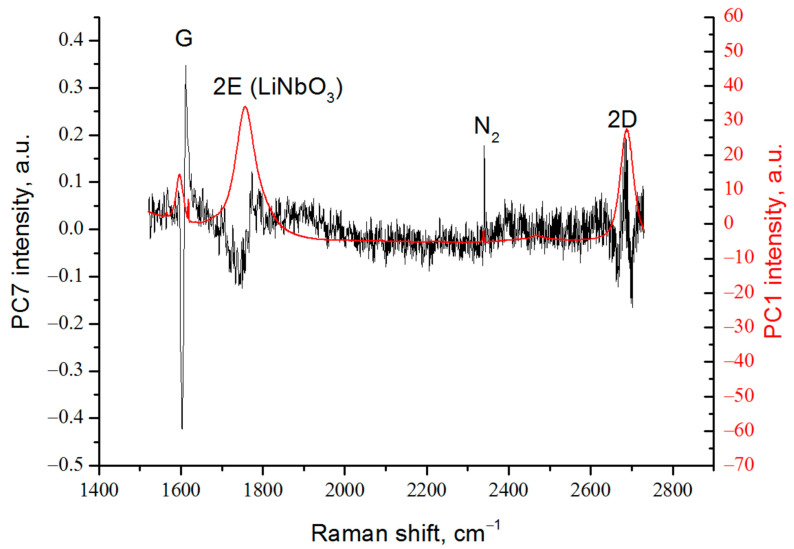
Comparison of load vectors PC1 and PC7.

**Table 1 nanomaterials-13-00350-t001:** Linear regression and correlation parameters of dependences from Figure 4.

	All Cycles	Cycle 1	Cycle 2	Cycle 3
Residual sum of squares	0.397	0.049	0.089	0.051
Correlation factor r	0.62	0.82	0.69	0.74
Intercept	−0.007	−0.071	−0.016	0.062
Slope	6.7 × 10^−5^	7.6 × 10^−5^	6.7 × 10^−5^	5.9 × 10^−5^

## Data Availability

All relevant data presented in the article are stored according to institutional requirements and as such are not available online. However, all data used in this manuscript can be made available upon request to the authors.

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
