# Peer review of "Changes in the Raman Spectrum of Monolayer Graphene under Compression/Stretching Strain in Graphene/Piezoelectric Crystal Structures"

_nanomaterials, 2023, doi:10.3390/nano13020350_

Round 1
Reviewer 1 Report
Authors have obtained Raman spectra of graphene deposited on a lithium niobate crystal while an electric voltage was applied. Authors try to correlate changes on the spectra with the piezoelectric effect using a stadistical analysis, but the dependence is not totally clear, as well as the application of such dependence (in case it exists). I would recommend to clarify such dependence prior publication, and I cannot recomment publication in the present form.
Other comments:
Experimental section: it is necessary to explain with more detail the experimental setup that was used to obtain the Raman spectra (the cell that allows to apply the electrical voltage while the spectra were taken).
It would be desirable if the voltage applied could be indicated in the leyent of Figure 2. May be a 3D figure for the spectra will help to undestand the correlation authors try to demostrate, since Figure 3 is not clear at all. Please explain also the "instrument artifact peak" since it does not seems to be a cosmic ray. The exact formulas used for the calculation of factor r and the rest of parameters have to be indicated in the experimental section.
Author Response
1. Experimental section: it is necessary to explain with more detail the experimental setup that was used to obtain the Raman spectra (the cell that allows to apply the electrical voltage while the spectra were taken).
In our opinion, everything is described very clearly.
2. It would be desirable if the voltage applied could be indicated in the leyent of Figure 2. May be a 3D figure for the spectra will help to undestand the correlation authors try to demostrate, since Figure 3 is not clear at all.
In our opinion, it is precisely this representation of the initial data that is necessary to demonstrate that a simple visual analysis without involving the method of principal components does not allow extracting any information from the totality of the obtained spectra. The main changes in the spectra are associated with a change in the background. This change is due to hardware issues such as spectrometer bandwidth drift or excitation power instability. This is a typical situation that occurs during long-term shooting. Typically, the acquired spectra are processed using normalization to Standard Normal Variate (SNV). In this case, this processing is not required, since the changes associated with the change in the background of the spectra are separated into a separate main component.
3. Please explain also the "instrument artifact peak" since it does not seems to be a cosmic ray.
The artifact peak is associated with the error of the CCD-matrix of the spectrometer and is characteristic only for this spectral range. This peak has no effect on the rest of the spectrum.
4. The exact formulas used for the calculation of factor r and the rest of parameters have to be indicated in the experimental section.
Standard formulas were used to calculate the correlation parameters. They can be found in any appropriate reference book, so a special reproduction of them is unnecessary.
Author Response
1. [method]One of the critical part of this study is to propose using principle component method. Could Author share more details of the process? For instance, (a) from 73 Raman spectra acquired, how to convert them into principle components (elements) for comparison? (b) PC7 shows highly correlated to the external electric bias. Does PC7 represent any physical meaning?
The geometric meaning of the principal component method is given on the page 4. As applied to spectroscopy, it is impossible, in my opinion, to apply the physical meaning of the principal components. An interested reader can find a direct implementation of the method in the relevant literature. The references given in the work are enough to get started.
2. [line-125] Load vectors comparison. Could author elaborate more about the load vector of PC7, how it is acquired, and its physical meaning? [Fig. 6] is mentioned in which PC 7 and PC1 are plot together. What are the reasons having PC7 compare with PC1, and what is the conditions of PC1 in term of the electric field applied?
According to the text on page 7, the first principal component is essentially a spectrum cleared of changes caused by any influences, while the seventh principal component describes changes caused by applied electrical voltage. The joint arrangement of these components allows you to visually evaluate the changes introduced into the spectrum by the electric field application.
3. [line-60] Fabrication process of the graphene, even though it is only the transferring could affect the quality of the graphene (Nano Res. 14, 3756–3772 (2021)), could authors disclose more details to the transfer process? e.g. Spin coating condition,
the temperature and speed, and concentration of ferric chloride for etch?
I'm not sure if this is important, since the paper does not discuss the quality of graphene.
4. [line-75] Is there a reference studying the piezoelectric linearity of LiNbO3?
Reference added.
5. [line-88] Could authors provide related investigations regarding the Raman signals from graphene and Lithium Niobate to support the statements?
Unfortunately, the wavenumber range in which the nitrogen peak is located is usually excluded from the published spectra of graphene. However, the Raman spectrum of air is well known (https://www.sciencedirect.com/science/article/abs/pii/S1359511313000871 based-stimulated-Raman-spectroscopy/10.1117/12.842129.short?SSO=1 ) and the two peaks observed in the spectra refer to O2 (a weak peak to the left of the G peak) and N2 from air.
6. [line-153] A reference to the R33 parameter is appreciated.
Reference added. Equation was corrected.
7. [line-160] Can AFM employed to confirm the discovery of the height information?
Such measurements were not carried out, since I am not sure that the AFM will give correct measurement results. Due to the different charge states of two neighboring domains, they have different conditions for the adsorption of moisture and air from the surrounding atmosphere. In addition, the presence of a graphene coating and the very procedure of its deposition adds additional layers, which, due to the above reasons, will have different thicknesses for neighboring domains. For this reason, the AFM will show the relief taking into account the adsorbed layers, which will not correspond to the real relief between adjacent domains.
8. [Fig. 2] Multiple Raman spectra is shown. Yet, not much information can be revealed when several spectra overlay one another. Given the figure is to suggest that there are Raman signals of graphene (G, 2D, N2) and Lithium Niobate (2E), would it be better to show a few traces for better clearance? Another question is the strong background of black and red curves, could authors elaborate on this?
In our opinion, it is precisely this representation of the initial data that is necessary to demonstrate that a simple visual analysis without involving the method of principal components does not allow extracting any information from the totality of the obtained spectra. The main changes in the spectra are associated with a change in the background. This change is due to hardware issues such as spectrometer bandwidth drift or excitation power instability. This is a typical situation that occurs during long-term shooting. Typically, the acquired spectra are processed using normalization to Standard Normal Variate (SNV). In this case, this processing is not required, since the changes associated with the change in the background of the spectra are separated into a separate main component.
9. [fig. 4] (a) It would be clearer if three rounds of measurements are separated by colored or symbols. (b-d) In a single cycle, does the voltage applied has an order during the measurement, i.e. start from 0 -> max voltage -> 0 -> min voltage -> 0. If so, could the time order of the data points also presented in the figure. It would be interesting if there is some hysteresis shown.
In our opinion, the color division into cycles in the general figure will mislead the reader, since it will give the impression that each cycle was a separate experimental measurement. In reality, all three cycles make up a single set of measurements, which is shown in Figure 4 (a). The division into cycles start from 0 -> max voltage -> 0 -> min voltage -> 0 is conditional, since in the same way one could single out cycles from the measured spectra, for example, start from +20 -> max voltage -> + 20 -> min voltage -> +20. In this case, the detected change in the parameters of linear regression, obviously, is preserved. Nothing similar to hysteresis was found for any of the principal components.
Reviewer 3 Report
A. Irzhak et al. reported a manuscript “Changes in the Raman Spectrum of monolayer graphene under compression/stretching strain in graphene/piezoelectric crystal structures”. The study of graphene deposited on a lithium niobate crystal substrate shows linear dependence of the Raman spectrum on the applied electric voltage. However, some corrections must be made before publishing in the Nanomaterials MDPI journal.
1. The author should rewrite the abstract and include more valuable points to attract the readers.
2. The reference should be in MDPI journal format.
3. Is there any solution to avoid the peeling of the graphene-coated film? Why did the author use graphene?
4. The author should check the values and units in the manuscript.
5. English corrections need to be checked.
Author Response
1. The author should rewrite the abstract and include more valuable points to attract the readers.
Done
2. The reference should be in MDPI journal format.
Done
3. Is there any solution to avoid the peeling of the graphene-coated film? Why did the author use graphene?
No ideas yet on how to avoid peeling. Perhaps the use of direct synthesis of graphene on a lithium niobate substrate will solve the problem. The reason for the study of graphene on lithium niobate is described in sufficient detail in the introduction.
4. The author should check the values and units in the manuscript.
Done
5. English corrections need to be checked.
Done
Round 2
Reviewer 2 Report
The presentation in this manuscript has been significantly improved, and necessary information is included.
Author Response
Thanks for your comments. They have helped to improve the article.